# A Simulation Experiment on Quality Dynamics of Reclaimed Water under Different Flow Exchanges

**DOI:** 10.3390/ijerph192013091

**Published:** 2022-10-12

**Authors:** Chengzhong Pan, Zhongfang Guo, Mingjie Luo

**Affiliations:** College of Water Sciences, Beijing Normal University, No. 19, Xinjiekouwai St., Haidian District, Beijing 100875, China

**Keywords:** degradation coefficient, reclaimed water, water exchange, microbial diversity

## Abstract

Reclaimed water plays an important role in maintaining urban aquatic ecosystems, especially in areas with water shortages. However, there is little information on water quality dynamics and its driving mechanism in reclaimed water bodies. The simulated experiments were conducted to investigate the effect of flow exchange on water quality dynamics and soil microbial diversity for 100% reclaimed water and mixed water (50% reclaimed and 50% stream water), and the exchange periods ranged from 2 to 40 days. The results showed that the degradation coefficients (*K*) of COD_Mn_ and NH_3_–N were 0.015 d^−1^ and 0.001 d^−1^ for the mixed water, while their *K* values were negative for the reclaimed water. The flow exchange had little effect on water quality dynamics for the mixed water, which may be attributed to the relatively low concentration of TP in this reclaimed water. A small or great exchange period led to a relatively high fluctuation in *K* during the experimental period and corresponded to a worse soil microbial diversity. These results indicate that it is not recommended to fill an isolated urban lake with 100% reclaimed water and that a suitable flow exchange period of 5~10 days could help self-purify the water quality.

## 1. Introduction

In the urban areas of North China, reclaimed water has been widely used to alleviate the stress of water shortage [1,2,3]. In Beijing, reclaimed water has accounted for proximately 30% of the total water resources consumption, and it was commonly used to replenish urban landscape water bodies [4,5,6]. Reclaimed water commonly has higher concentrations of COD, TN, and TP than naturally fresh water, so it brings about a higher risk of the outbreak of algal blooms [7,8]. Water quality security has not only affected the diversity and stability of aquatic ecosystems but is also related closely to the health of the surrounding residents [9]. Therefore, it is urgent for us to properly use limited reclaimed water resources to raise the urban water environment systems.

Laboratory experiments, field investigations and mathematical models were frequently used to investigate water quality dynamics in river channels or lakes [10]. Much research showed that flow or water exchange played an important role in water quality dynamics. For instance, Pei and Ma (2002) established a mathematical model in predicting the effect of the water exchange period on the water quality of West Lake in Hangzhou, China, and they found that the concentrations of several common pollutants decreased with the increasing water diversion rates [11]. He et al. (2016) resorted to the principle of a fully mixed reactor and simulated the effect of the exchange period on algae growth and showed that a rapid water exchange tended to trigger greater flow turbulence in the water body and damaged the formation of algae blooms [12]. Although much research showed that a rapid flow fluidity could increase the self-purification capacity of pollutants [13,14], there is little information on water quality dynamics for reclaimed water bodies with different flow exchanges using contrast experiments.

Additionally, Water quality dynamics were not only affected by flow or water exchange but also by soil or plant microbial communities in water bodies [15]. In an urban lake replenished by reclaimed water, the proportion of reclaimed water and its flow exchange is extremely important to sustain the maintain the sustainable water environmental system. So far there is limited information on pollutant degradation and its impacting mechanism, especially the interaction between water quality and soil or sediment microbial diversity [6,15,16,17].

In this study, a simulated experiment was conducted to investigate the effect of the exchange period on water quality dynamics, and the 100% reclaimed water and mixed water with different inflow exchanges were designed to comparatively analyze the differences in self-purification processes of water quality and soil microbial diversity. The main objectives of this study are (1) to clarify the effect of the exchange period on water quality dynamics and soil microbial diversity, (2) to understand the mechanism of reclaimed water quality dynamics, and (3) to propose a suitable a replenishment flow rate or exchange period. These results will be helpful to allocate and use reclaimed water to improve urban surface water systems.

## 2. Methods

### 2.1. Experimental Apparatus

The Yongding river flows into the urban or skirt area of Beijing, and it played a very important role in the social and economic development in Beijing. However, since 1980, the upstream runoff has been greatly decreasing due to climate change and the unreasonable exploitation and use of water resources.

Meanwhile, the river segment located in the urban area of Beijing had a very strong leakage capacity due to the sandy riverbed. In the 2000s, in order to restore the landscape of the water body, the “Five Lakes and One Line” engineering project in the plain section of the Yongding River was constructed. The project consists of five typical shallow lakes which can be connected by a water channel if no replenishment runoff occurs.

Urban reclaimed water is considered an important replenishment resource for the landscape lakes along the Yongding river. However, it is still a pending question on how much and how reclaimed water flows into the lakes.

Big buckets were used to simulate the isolated urban lakes, and two types of water were introduced into the buckets to investigate the water quality dynamics. One is 100% reclaimed water (reclaimed water) and the other is 50% reclaimed water and 50% upstream water (mixed water). The water in two buckets was circulated by two circulating pumps (Figure 1). The capacity of each bucket is 60 L, and two large buckets were used to simulate a lake. The influence of flow rate or exchange period on water quality changes was studied. The five treatments in circulation flow rates were designed as 0, 7, 14, 21, and 28 L/d, and they corresponded to 40, 17.1 d, 8.6 d, 5.7 d, and 4.3 d. The “0” circulation rate represented the static water body without water circulation or exchange.

The circulating pumps used were the Kamoer DIP intelligent peristaltic pump model no. 50-b253. The circulating flow rate was adjusted by the waves of the pumps. Water quality indicators were measured using conventional methods (Table 1).

The reclaimed water used in the experiment was taken from Beijing Lunan Sewage Treatment Co., Ltd. (Beijing, China), which is the main water resource in the projects of “Five lakes and one line”. The upstream runoff mainly came from the Guanting reservoir of the Yongding River, and the runoff rates had been decreasing trend in recent years.

The first 10 cm of bottom soil was collected from the riparian zone of the Yongding River bank using a mud dredge to investigate the effect of soil microbial community on water quality changes. The riparian had a smaller soil particle size than the river bed, and silt and clay particles approximately accounted for 65% of the total soil. The collected soil was placed in an incubator and immediately brought back to the laboratory. The soils were mixed enough to ensure the same soil microbial community, and they were separately filled into 30 cm × 40 cm gauze bags with 1 kg. One tested bucket corresponded to a bucket bag, and 20 bags and buckets were used in this experiment. The initial water quality of reclaimed water and mixed water is shown in Table 2.

### 2.2. Experimental Processes

The experiment lasted for 40 days. Water quality indexes, including water temperature, pH, dissolved oxygen, ammonia nitrogen, REDOX potential, COD, and electrical conductivity were regularly tested in each group. The average value of each index was obtained by repeating the measurement three times for each sample. The experiment was able to investigate changes in the water quality and soil microbial community over long water exchange periods.

(1)Measurement of water quality dynamics

Ammonia nitrogen and COD were measured every 8 days and analyzed graphically. Water temperature, pH, dissolved oxygen, REDOX potential, and conductivity water quality indicators were measured at 7 a.m. every day. The water pump was run until the flow was stable. The experiment period commenced when stable flow had been achieved.

(2)Soil microbial community

Soil microbes in the riparian zone may be crucial to the degradation of pollutants in aquatic ecosystems. In this study, the soil samples taken from the urban river bank were suspended inside buckets with different water exchange periods to simulate the impact of soil microbial communities on water quality dynamics. The interactions between soil microbial community and water quality dynamics were investigated for different treatments. Before and after the experiment, 20 g of soil was collected to measure soil microbial community indicators.

All the DNA analyses of samples were completed by a professional gene sequencing company, Shanghai Meiji Biomedical Technology Co., Ltd. (Shanghai, China). The soil microbial community was analyzed on the Illumina HiSeq and MiSeq platforms, and the pair reads were overlapped and filtered using a Trimmomatic system.

### 2.3. Pollutant Degradation Coefficient

The degradation process of pollutants in water can be expressed using first-order reaction kinetics:(1)$$\frac{dC}{dt}=-KC$$
where *t* is the reaction time (d), *K* is the comprehensive degradation coefficient of pollutants (d^−1^), *C* is the pollutant concentration measured at time *t* (mg/L), and *C*_0_ is the initial pollutant concentration (mg/L), so the integral is
(2)$$C=C_{0}e^{-Kt}$$

The calculation formula of degradation coefficient *K* is
(3)$$K=\frac{1}{t}ln\left|\frac{C_{0}}{C}\right|$$

All the tests were conducted in the indoor laboratory, and each treatment was subjected to the same air temperature and humidity conditions. Therefore, the *K* values of each treatment should be comparable [18,19].

### 2.4. Soil Microbial Community Richness and Diversity

In this study, microbial alpha diversity reflects the microbial diversity within a specific region or ecosystem. The dilution curve, community richness including Chao and Ace estimators, community diversity including Shannon and Simpson indexes, and the Good’s coverage were calculated based on the number of individuals randomly selected from the sample, and the number of all species. It can be used to compare the species richness of samples with different sequencing data amounts and to explain whether the sequencing analysis of samples is reasonable. The Chao and ACE estimators can reflect the community richness, and they are used to evaluate the number of OTU in a sample. A larger Chao index indicates that there are more species in this sample.

The Good coverage can be calculated using Formula (4):(4)$$Coverage=1-\frac{n_{1}}{N}$$
in which *n*_1_ is singletons referring to the number of OUT in a single sequence, and *N* is the number of all sequences that occurred in the sample. The index reflects the reality of the sequencing results. The higher the coverage value is, the higher the probability that the sequence in the sample is detected, which means a trustable test.

The Shannon and Simpson indexes in community diversity were calculated for the soil samples, and they can be expressed as Formulas (5) and (6):(5)$$Shannon=-\sum_{i=1}^{Sobs}\frac{n_{i}}{N}ln\left(\frac{n_{i}}{N}\right)$$
(6)$$Simpson=\frac{\sum_{i=1}^{Sobs}n_{i}\left(n_{i}-1\right)}{N\left(N-1\right)}$$
in which *n_i_* is referring to the number of OUT in a single sequence, sobs is the observed number of OUT in a sample. The larger the Shannon and Simpson values are, the higher the community diversity is.

## 3. Results and Analysis

### 3.1. Pollutant Concentration Dynamics under Different Exchange Flow Regimes

The reclaimed water had a significantly higher concentration in TN than the mixed water. However, the mixed water even had greater concentrations in COD_Mn_, NH_3_–N, and TP than the reclaimed water (Figure 2).

As the experimental days increased, the concentrations of TN and TP greatly decreased, especially during the period of 10–15 days. This trend may indicate that the environmental conditions and soil microbial community would help the degradation of TN and TP. The reclaimed water corresponded to an increasing trend in COD, which differed from the mixed water (Figure 2a,a’). The concentration in NH_3_–N had a greater fluctuation than that in other indicators. It may hint that NH_3_–N may be more sensitive to environmental conditions than TN and TP.

In this experiment, the exchange flow discharge was assigned from 0 to 28 L/d, corresponding to the exchange period of 4.3 d to 40 d. The greatest exchange flow rate of 28 L d^−1^ generated a relatively small concentration at the final phase of experiments (40th days) (Figure 2). However, the ANOVA showed that the exchange flow regime had no significant effect on the concentration dynamics.

### 3.2. The Effect of Water Exchange Period on Degradation Coeffcicent

Based on the concentration difference between the original and final phase of each test, the degradation coefficients for different water quality indexes were calculated by Formula (3). For both reclaimed and mixed water, the degradation coefficients (*K*) in TN and TP appeared positive and their concentrations decreased with experimental days (Figure 2 and Figure 3). Comparing the two tested water, the *K* values in TN varied from 0.025 to 0.037 for the reclaimed water, and from 0.003 to 0.008 for the mixed water. The reclaimed water had a significantly greater *K* value in TN than the mixed water (Figure 3c); for both tested waters, the *K* values in TP ranged from 0.012 to 0.035, and the mixed water generated a relatively greater *K* in TP value than reclaimed water under the exchange period of 5.7–17.1 days.

Surprisingly, for reclaimed water, COD and NH_3_–N generated negative degradation coefficients for all exchange periods (Figure 3a,b). These phenomena may be drawn from the increasing concentrations with experimental days (Figure 2). However, for mixed water, the concentrations in COD and NH_3_–N generally decreased with the increasing experimental days, and it generated positive *K* values (Figure 3). Therefore, from the perspective of the degradation in COD and NH_3_–N, 100% reclaimed water should not be recommended to replenish the isolated urban lakes.

For the reclaimed water, the water exchange had little effect on the degradation coefficients in COD and TP, but a significant effect on NH_3_–N and TN. For NH_3_–N, the smallest exchange period of 4.3 d corresponded to the greatest *K* values. It indicates that for isolated urban lakes replenished with reclaimed water, the frequent water exchange may help to raise the self-purification of water quality.

Figure 4 showed that the degradation coefficients in COD_Mn_, NH_3_–N, TN and TP varied with the experimental period for reclaimed and mixed water. Generally, the reclaimed water had a greater variation range in *K* than the mixed water. For instance, the *K* values in NH_3_–N ranged from −0.25 to 0.25 for reclaimed water, and those varied from −0.05 to 0.05 for mixed water. This result hints that the degradation of pollutants in reclaimed water was more sensitive to environmental conditions (e.g., air temperature, humidity) than those in mixed water.

For both reclaimed and mixed water, there were greater *K* values at the beginning of the experiment than those at the latter phases. In other words, the *K* values had a decreasing trend with the increasing experimental periods. For COD_Mn_ and NH_3_–N in mixed water, the *K* values gradually decreased from positive to negative (Figure 4a’,b’). For other pollutants, the variation in *K* values in the experimental phase had great fluctuation (Figure 4). It indicates that the self-purification of water quality in an isolated surface lake will decrease with the increasing retention period. It highlights the importance of flow exchange.

The high or small flow rates tended to correspond to the relatively great fluctuations in degradation coefficients during the experimental period. Nevertheless, the effect may be more complicated and can be closely dependent on other environmental conditions. Consequently, the flow exchange flow rates or periods had no clear impact trend on the dynamics of degradation coefficients during the different experimental phases.

### 3.3. The Effect of the Water Exchange Period on Soil Microbial Communities

The soil microbial communities before and after each test for different treatments were compared to further discuss the effect of flow exchange on water quality dynamics.

An operational taxonomic unit (OTU) is used to represent a taxon (strain, species, genus, grouping, etc.) in phylogeny or population genetics, and sobs are the number of OTUs measured. A total of 727,595 sequences were obtained using high-throughput sequencing and noise reduction of the 16SrRNA gene 454 from the soil DNA in the two groups. The sequence number of each sample ranged from 62,035 to 70,603. The operable taxa at the species level were calculated using the criteria of 97% similarity. Consequently, a dilution curve was drawn by random sampling of the sequence, and a rationality analysis of the sequencing data volume was performed in Figure 5. 

The current sequencing depth could not totally cover or saturate the abundance of the microbial community in the collected soil samples, but most of the bacteria should have been collected. The current curve appears relatively smooth, and the additional data can only produce limited new species [20].

For mixed water, the sobs index of soil species richness increased with the increasing exchange flow discharges (Figure 5b). For reclaimed water, the smallest and highest flow exchange flow rates (0 and 28 L d^−1^) had a similar dilution curve, and the medium exchange flow rates including 7–21 L d^−1^ had a better performance in species richness than a high exchange flow rate of 28 L d^−1^ (Figure 5a). However, for mixed water, except for the blank exchange flow rate, the species richness increased with the increasing water exchange flow rates (Figure 5b). The difference in soil dilution curve between mixed and reclaimed water hints that the low or high exchange flow rates may do damage to soil microbial activity in the aquatic ecosystem, and the reclaimed water may bring about a positive relationship between soil microbial activity and flow exchange or connectivity [21]. Meanwhile, the low and high exchange flow rates of 0 and 28 L d^−1^) have great variability in the sobs index and in *K* values (Figure 4 and Figure 5). Their consistency further highlights the close relationship between soil microbes and the degradation of water pollutants.

Table 3 showed the main soil microbial richness and uniformity indexes in the reclaimed and mixed water with different exchange flow rates. For all tests, the Good’s coverages of soil samples reached 0.99, which indicated that the detection data on the soil microbial community were well enough to analyze the diversity and richness.

Sobs ranged from 35 to 43 for reclaimed water, and from 38 to 42 for mixed water. Compared with the sobs value of initial soil (40), the medium exchange flow rates of 7–21 L d^−1^ had greater sobs values and performed better than the smallest or greatest flow rates (0 and 28 L d^−1^) (Table 3).

The Chao and Ace indexes mainly focus on the number of species in a soil community. For reclaimed and mixed water, the Chao and Ace indexes ranged from 37 to 46, and there was no significant difference at *p* = 0.05 level between the two tested waters using the paired t-test. As the exchange flow rates increased from 0 to 28 L d^−1^, there was no obvious changing trend for Chao and Ace indexes.

In terms of community diversity, a paired t-test showed that there was no significant difference in Shannon and Simpson indexes between reclaimed and mixed water. Most of the different exchange flow rates had lower diversity indexes compared with the initial soil. The above results hints that soil microbial community diversity has a decreasing trend in an isolated water body even if there exists an internal water exchange or movement.

## 4. Discussion

### 4.1. Reclaimed Water Use

In terms of the reuse of urban reclaimed water, the European Union, the United States, and other developed countries have imposed strict limits on the amounts of nitrogen and phosphorus that may replenish urban surface water bodies. These limits proposed a higher treatment standard for reclaimed water or urban sewage [2,22]. Reclaimed water is strictly treated and deeply purified through activated carbon filters and the ozone disinfection process before it is used to replenish urban lakes or rivers [3,23]. Since reclaimed water has relatively high nitrogen and phosphorus contents than natural river runoff, it is necessary to reduce their contents to control the occurrence of eutrophication, especially in isolated water bodies [24,25].

In Beijing, reclaimed water has been playing an important role in providing urban lakes [5]. Additionally, some urban lakes replenished with reclaimed water tended to be isolated from the natural river system. Consequently, some eco-environmental problems including water quality deterioration and eutrophication occurred. The water internal exchange was regarded as a feasible solution to resolve the above problems since there was very limited upstream runoff.

This study found that for 100% reclaimed water, the concentration of COD_Mn_, NH_3_–N continuously increased during the experimental period, and there were negative degradation coefficients. However, for mixed water, the concentration of COD_Mn_, NH_3_–N had a decreasing trend. These results indicate that urban isolated waters should only hold a suitable proportion of reclaimed water, and they should not be filled with 100% reclaimed water. Other researchers also suggested a suitable proportion of reclaimed water. For instance, He et al. (2016) found that the water quality deterioration in isolated ponds accelerated when the proportion of reclaimed water was equal to 75% and 50% [12]. Wang (2019) took accounts into the dynamics of both water quality and soil microbial diversity and suggested the reclaimed water should not exceed 75% in an isolated lake [26]. Liu (2019) suggested that the reclaimed water should not exceed 50% of urban landscape water bodies [13]. Therefore, the proportion of reclaimed water should be less than 50% in urban lakes.

### 4.2. Water Exchange and Degradation Coefficient

When isolated urban lakes were filled with reclaimed water, the water exchange period can be an adjustable important parameter that may affect the concentration and residence time of pollutants and nutrients, as well as the biological and chemical reactions [9,24,27]. For both the two tested waters, a small or great water exchange period tended to have a great fluctuation in degradation coefficients during the experimental period (Figure 5), and the exchange water periods ranging from 5.7 to 17.1 days had better performance in self-purifying the typical water pollutants.

In this study, for reclaimed water, the degradation coefficients of TN and TP had increasing trends with the increasing exchange flow rates (Figure 4). However, for COD_Mn_, NH_3_–N, the exchange period had no significant effect on their degradation coefficients for both reclaimed water and mixed water. Ao et al. (2018) and Liu (2019) found that in urban ponds mainly replenished by reclaimed water, the N and P content in water bodies decreased with the decreasing water exchange periods. In this study, the exchange period had little impact on the N and P content [13,28]. The difference in our study may be attributed to the relatively smaller TP content in the reclaimed water. The high ratio of N to P and the small P content may limit the removal processes of nutrients in water bodies. These results need further clarification based on more experiments and investigations under different environmental conditions.

Similar to the relationship between water exchange and degradation coefficients, for both reclaimed and mixed water, the medium exchange rates of 7 to 21 L d^−1^ corresponding to the periods of 5.7 to 17.1 days had a better performance in soil micro-community richness and diversity than a small and great exchange flow rate (Table 3). It is speculated that quiet or frequently disturbed water could not generate more types and quantities of microorganisms for both the reclaimed and mixed water. Further, the poor soil micro conditions may result in the smaller degradation coefficients in these organic pollutants [26]. The above results may hint that 5.7~17.1 days of exchange period would be recommended to use part of reclaimed water to replenish an urban lake.

Liu (2019) also conducted a series of experiments on water quality dynamics and reclaimed water exchange periods and found that the water quality deterioration accelerated in summer than in spring, and the exchange period in urban lakes should be 3 days in summer, and 5 days in spring and autumn [13]. Our study suggested that frequent water exchange could not be recommended since it may bring about a decrease in soil microbial diversity. Additionally, a fast water exchange means more energy consumption. The result from this experiment highlights the importance of a suitable exchange period (e.g., 5–10 days) in water quality purification [12,26].

### 4.3. Limitations of This Study

This study used laboratory simulations to probe into water quality processes in urban lakes replenished by reclaimed water, and their response to water exchange. The degradation coefficients and soil microbial community were carefully investigated for the reclaimed and mixed water under experimental conditions. However, the buckets used in this study were not big enough to represent a real urban lake, and the indoor experimental conditions clearly differed from the natural environmental conditions. The difference in environmental factors could have a negative influence on the extrapolation of this research. Meanwhile, the relatively short experimental period of 40 days could not cover the whole dynamic processes of water quality and soil micro-community.

Nevertheless, it is very difficult to carry out this investigation in an urban lake with different water sources and water exchange flow rates. This study paid more attention to the tendency in the degradation coefficient with the water exchange period and attempted to find a suitable exchange rate, which can help to operate and manage isolated urban lakes replenished with reclaimed water. Moreover, the study discussed the dynamics of several typical pollutants including COD_Mn_, NH_3_–N, TN and TP, and the effects of soil macrobiotics on water quality. In the future, more organic pollutants and poisonous indexes, and environmental conditions should be considered that can be more sensitive to the health of aquatic species in the replenishment of rivers and lakes; it would be better to conduct an on-site and long-term investigation using natural ponds or lakes.

## 5. Conclusions

To better understand the effect of water exchange on water quality dynamics in urban lakes, a laboratory simulation experiment was conducted using 100% reclaimed water and mixed water (50% reclaimed water and 50% river water) during a 40-day experimental period. The degradation coefficient (*K*) of several typical pollutants including COD_Mn_, NH_3_–N, TN and TP, and soil microbial alpha diversity in the two groups of experiments were comparably analyzed and discussed. For reclaimed water, the degradation coefficients of TN and TP generally decreased with the increasing water exchange period. However, the *K* values of COD_Mn_ and NH_3_–N had little relation to water exchange. Moreover, the reclaimed water had negative *K* values in COD_Mn_ and NH_3_–N under all water exchanges, while the mixed water appeared to have a positive self-purification impact. This implies that it is not recommended to replenish using 100% reclaimed water in an isolated urban lake.

For both reclaimed and mixed water, a small or great exchange flow rate led to a greater fluctuation in the degradation coefficient during the experimental period. Meanwhile, the medium exchange periods ranging from 5.7 to 17.1 days had a better performance in soil micro-community richness and diversity than the small or great water exchange did. The water exchange period could be taken into account to manage urban lakes mainly replenished with reclaimed water. These results can provide guidance to effectively use reclaimed water and sustain an urban isolated aquatic ecosystem. It is worth further investigating the water quality dynamics and its impact mechanism under the conditions of combined natural water ecosystems with adjustable impact factors, including the proportion of reclaimed water and its sensitive environmental conditions.

## Figures and Tables

**Figure 1 ijerph-19-13091-f001:**
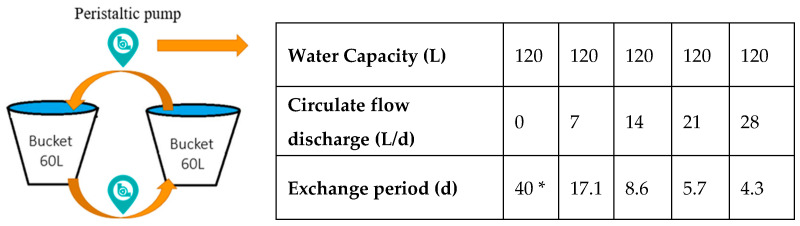
Experiment setup and flow exchanges. * refers to the experimental length.

**Figure 2 ijerph-19-13091-f002:**
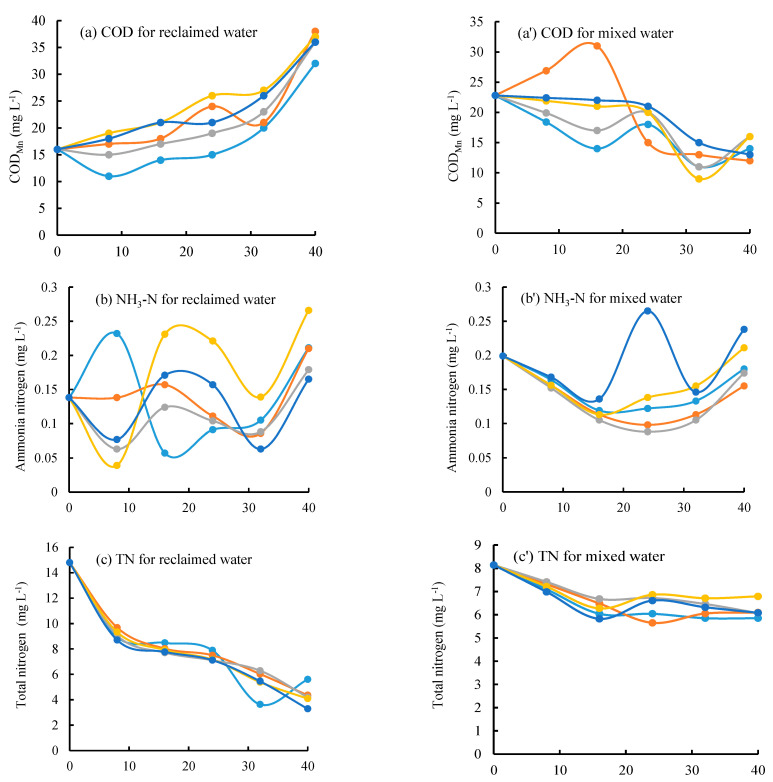
Temporal variations in COD_Mn_, NH3–N, TN and TP concentrations during the experimental period (40 days). For different exchange flow rates ((**a**–**d**) for the reclaimed water and (**a’**–**d’**) for mixed water).

**Figure 3 ijerph-19-13091-f003:**
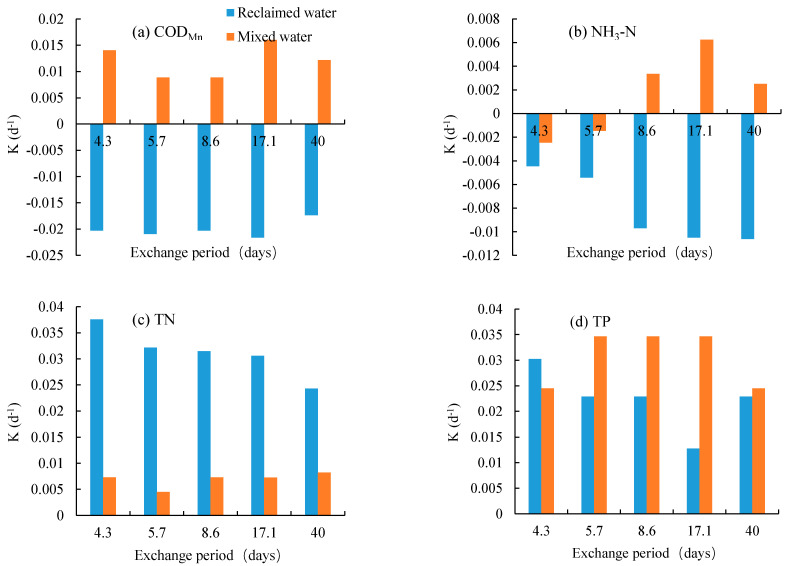
The degradation coefficients (*K*) change in COD_Mn_, NH_3_–N, TN and TP with water exchange periods. for reclaimed and mixed water. (**a**–**d**) represent four typical pollutants.

**Figure 4 ijerph-19-13091-f004:**
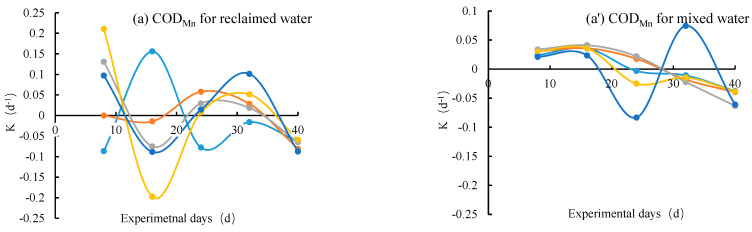
Dynamics of degradation coefficient (*K*) in CODMn, NH3–N, TN and TP with the experimental processes. for different exchange flow rates ((**a**–**d**) for the reclaimed water and (**a’**–**d’**) for mixed water).

**Figure 5 ijerph-19-13091-f005:**
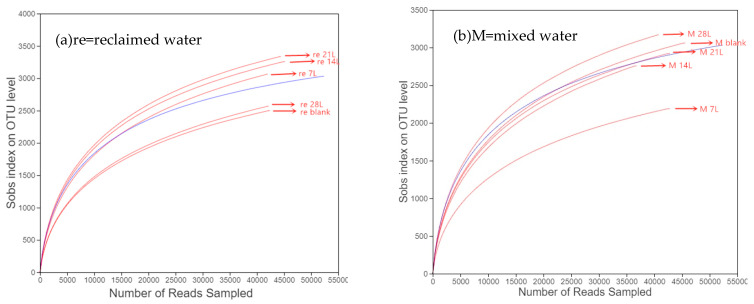
Dilution curves for soil communities in the reclaimed (**a**) and mixed water (**b**). with different exchange flow rates (the blue curve represents the soil before tests).

**Table 1 ijerph-19-13091-t001:** Measuring methods of typical water quality indexes.

Water Quality Indicators	Determination Method
pH	PHscan10S Pen type pH meter
Dissolved oxygen (DO)	Dissolved oxygen meter DO200A
Chemical oxygen demand (COD_Mn_)	Automatic COD_Mn_ analyzer
Ammonia nitrogen(NH_3_–N)	Sodium reagent spectrophotometry
Total nitrogen (TN)	Titrimetric method after distillation
Total Phosphorus (TP)	Ammonium Molybdate Spectrophotometric Method
Electrical conductivity	2265FS Portable conductivity meter

**Table 2 ijerph-19-13091-t002:** Initial water quality of the two water bodies.

Water Quality Index	Reclaimed Water	Mixed Water
pH	8.6	8.9
Dissolved oxygen (mg/L)	4.12	4.98
Ammonia nitrogen (mg/L)	0.36	0.03
Nitrate nitrogen (mg/L)	13.2	4.4
Nitrite nitrogen (mg/L)	0.03	0.03
Chemical oxygen demand (COD_Mn_) (mg/L)	15	34
Electrical conductivity (mS/cm)	1.04	0.86

**Table 3 ijerph-19-13091-t003:** Soil alpha diversity indexes in the different water exchange flow rates (EFR) for reclaimed and mixed water.

	Treatments (EFR)	Sequences	Sobs	Shannon	Simpson	Ace	Chao	Coverage
	Initial soil	52,337.00	40.00	2.10	0.21	40.35	40.00	0.99
	0 (blank)	42,166.00	35.00	1.98	0.20	39.44	37.00	0.99
Re	7 L d^−1^	41,923.00	42.00	1.91	0.25	45.28	43.50	0.99
	14 L d^−1^	44,819.00	43.00	2.23	0.16	44.77	43.60	0.99
	21 L d^−1^	44,241.00	41.00	2.05	0.21	42.95	42.50	0.99
	28 L d^−1^	42,071.00	36.00	1.87	0.23	46.01	38.00	0.99
	0 (blank)	45,228.00	38.00	2.05	0.21	44.80	41.00	0.99
	7 L d^−1^	42,423.00	39.00	1.93	0.20	41.83	42.00	0.99
Mix	14 L d^−1^	36,502.00	39.00	1.71	0.31	40.65	40.50	0.99
	21 L d^−1^	42,777.00	42.00	1.95	0.23	44.35	42.75	0.99
	28 L d^−1^	40,726.00	39.00	2.15	0.19	39.33	39.00	0.99

## Data Availability

Not applicable.

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
