# Peer review of "A Simulation Experiment on Quality Dynamics of Reclaimed Water under Different Flow Exchanges"

_ijerph, 2022, doi:10.3390/ijerph192013091_

Round 1

Reviewer 1 Report (Previous Reviewer 1)

The study examines the effect of flow exchange on water quality changes in reclaimed water replenishment city lakes. Remarks: How the results of the present work are expected to influence potential readers? This should be discussed in the point concerning the discussion of the results. Please present in the abstract the contribution of the paper to the literature and the outline of the paper. The advantages and novelty of the research approach need to add. This will help highlight any unique findings. 

Author Response

Reviewer #1:

The study examines the effect of flow exchange on water quality changes in reclaimed water replenishment city lakes.

Remarks: How the results of the present work are expected to influence potential readers? This should be discussed in the point concerning the discussion of the results. Please present in the abstract the contribution of the paper to the literature and the outline of the paper. The advantages and novelty of the research approach need to add. This will help highlight any unique findings. 

RE:Thanks for the reviewer’s precious suggestions. We have supplemented three discussions in the 4. Discussion (see 4.1 and 4.1). The discussion mainly focused on 1) the reclaimed water proportion to urban lakes, 2) the difference in water quality purification from other researches; 3) the impact of exchange period on water quality dynamics.

Meanwhile, we have supplemented some references and revised the abstract. We have greatly revised the section of introduction, and supplemented some contents on the novelty and advantages of this study. The added content on novelty and advantage are:

“Although many researches showed that a rapid flow fluidity could increase the self-purification capacity of pollutants (Liu, 2019; Ćmiel et al., 2020), there is little in-formation on water quality dynamics for reclaimed water bodies with different flow exchanges using contrast experiments.” (see p2 line44-47).

“So far there are limited information on pollutant degradation and its impacting mech-anism, especially the interaction between water quality and soil or sediment microbial diversity (Liu, 2019; Kong & Koelmans, 2019; Chang etal., 2020).” (see p2 line 52-54)

“The main objectives of this study are 1) to clarify the effect of exchange period on wa-ter quality dynamics and soil microbial diversity, 2) to understand the mechanism of reclaimed water quality dynamics, and 3) to propose a suitable a replenishment flow rate or exchange period.” (see p2 line 58-61)

Reviewer 2 Report (New Reviewer)

This study reported the use of a laboratory simulation experiments to analyze the water exchange process and underlying mechanisms affecting the water ecological environment of urban lakes. Multiple physicochemical properties of reclaimed water and mixed water were measured and the in-depth analysis was performed to deduce the conclusions. The reviewer feels that it can be considered for publication after considering the following moderate comments:

(1) The major concern is that the measured values in the tables and figures have no standard errors/deviations, which are very important for scientific research.

(2) The Introduction section can be improved by clearly signifying the research gap and environmental implications of this work. The references should also be updated with the latest ones (from 2018-2022).

(3) More in-depth mechanistic discussions are needed to compare the current results with the literature.

Author Response

Reviewer #2:

This study reported the use of a laboratory simulation experiments to analyze the water exchange process and underlying mechanisms affecting the water ecological environment of urban lakes. Multiple physicochemical properties of reclaimed water and mixed water were measured and the in-depth analysis was performed to deduce the conclusions. The reviewer feels that it can be considered for publication after considering the following moderate comments:

Re: thanks very much for the reviewer’s high evaluation. According to the author’s comments and suggestions, we have carefully revised and supplemented the manuscript.

  • The major concern is that the measured values in the tables and figures have no standard errors/deviations, which are very important for scientific research.

Re: Thanks for the reviewer’s suggestions. During the experiment processes, each week, we took one water and soil sample to measure the water and soil quality indexes. For some indexes, one sample was analyzed with three replicates using different instruments, but there is a very small error for the three replicates due to the same sample. And we just used the average value to draw these figures. For some indexes, we just got one measurement data for each sample, so we had to use the data to analyze results. In the following research, we will take more samples and reduce the sampling volume, and can show the error bar according to the reviewer’s suggestion.  

(2) The Introduction section can be improved by clearly signifying the research gap and environmental implications of this work. The references should also be updated with the latest ones (from 2018-2022).

Re: According to the reviewer’s suggestions, we have greatly revised the section of introduction, and supplemented some contents on the novelty and advantages of this study. The added content on novelty and advantage are:

“there is little information on water quality dynamics for reclaimed water bodies with different flow exchanges using the method of comparative investigations.”(see p2 line47-49).

“For reclaimed water bodies, there are limited information on pollutant degradation and its impacting mechanism, especially the interaction between water quality and soil or sediment microbial diversity (Liu, 2019; Kong & Koelmans, 2019; Chang etal., 2020).” (see p2 line 55-58)

“The objective of this study is 1) to clarify the effect of exchange period on water quality dynamics and soil microbial diversity, 2) to understand the mechanism of reclaimed water quality dynamics, and 3) to propose a suitable a replenishment flow rate or exchange period.” (see p2 line 62-64)

Meanwhile, we further improved the references, and added some latest references:

Cherchi, C., Kesaano, M., Badruzzaman, M., Schwab, K., Jacangelo, J.G., Municipal reclaimed water for multi-purpose applications in the power sector: a review. J. Environ. Manag. 2019, 236, 561–570. https://doi.org/10.1016/j.jenvman.2018.10.102.

Li, Q., Wang, WJ; Jiang XH, Lu, DL; Zhang, YB, Li JX, Analysis of the potential of reclaimed water utilization in typical inland cities in northwest China via system dynamics, Journal of Environmental Management, 2020, 110878

Du, Y., Lv, X.T., Wu, Q.Y., Zhang, D.Y., Zhou, Y.T., Peng, L., Hu, H.Y., 2017. Formation and control of disinfection byproducts and toxicity during reclaimed water chlorination: a review. J. Environ. Sci. 58, 51–63. https://doi.org/10.1016/j.jes.2017.01.013.

Chang NN; Zhang QH; Wang Q; Luo, L; Wang XC; Xiong JQ; Han JX. Current status and characteristics of urban landscape lakes in China, Science of the Total Environment, 2020, 712, 135669.

Deng, S., Yan, X., Zhu, Q., Liao, C., The utilization of reclaimed water: possible risks arising from waterborne contaminants. Contaminants. Env., 2019, 254, 113020.

Liu YZ. Study on water quality characteristics and regulation technology of urban landscape water body under reclaimed water recharge condition, Xi'An University of Architecture and technology, Xi’an, China, 2019.

Ao D, Chen R, Wang XC, Liu YZ, Dzakpasu M, Zhang L, Huang Y, Xue T, Wang N. On the risks from sediment and overlying water by replenishing urban landscape ponds with reclaimed wastewater. Environmental Pollution. 2018, 236, 488-497.

He T, Xiong JQ; Wang XC; Liu YZ; Quality variations of landscape water with different ratio of reclaimed water supply. Chinese Journal of Environmental Engineering, 2016,10(12):6923-6927. (in Chinese)

Wang XY; Pan CZ; Liu CL; Guo ZF. Experimental research on water quality evolution mechanism of landscape water bodies with different ratios of reclaimed water. South-to-North Water Transfers and water Sciences & Techenology, 2019,17(1): 76-83 (in Chinese)

(3) More in-depth mechanistic discussions are needed to compare the current results with the literature.

Re: According to the reviewers suggestion, we have supplemented some discussions in the 4. Discussion 4.1 and 4.2. The details are as follows:

“Other researchers also suggested a suitable proportion of reclaimed water. For instance, He (2016) found that the water quality deterioration in isolate ponds accelerated when the proportion of reclaimed water were equal to 75% and 50%. Wang (2019) took ac-counts into the dynamics of both water quality and soil microbial diversity, and sug-gested the reclaimed water should not exceed 75% in an isolated lake. Liu (2019) sug-gested that the reclaimed water should not exceed 50% of urban landscape water body. Therefore, it should be better that the proportion of reclaimed water can be less than 50% in urban lakes.” (see p11 line 322-339)

 “Ao (2018) and Liu (2019) found that in urban ponds mainly replenished by reclaimed water, the N and P content in water bodies decreased with the decreasing water ex-change periods. In this study, the exchange period had little impact on the N and P content. The difference from our study may be attributed to the relatively smaller TP content in the reclaimed water. The high ratio of N to P and the small P content may limit the removal processes of nutrients in water bodies.” (see p12 line 351-356)

“Liu (2019) also conducted series of experiments on water quality dynamics and reclaimed water exchange periods, and found that the water quality deterioration ac-celerated in summer than in spring, and the exchange period in urban lakes should be 3 days in summer, and 5 days in spring and autumn. Our study suggested that the frequent water exchange could not be recommended since it may bring about a de-crease in soil microbial diversity. Additionally, a fast water exchange means more en-ergy consumption. The results from this experiment highlights the importance of a suitable exchange period (e.g. 5-10 days) in water quality purification (He, 2016; Wang et al., 2020).” (see p12 line 368-375)

Round 2

Reviewer 1 Report (Previous Reviewer 1)

The study examines the effect of flow exchange on water quality changes in reclaimed water replenishment city lakes. Reclaimed water is an importance supplement to water resources in the areas of water shortage, and it plays an important role in maintaining aquatic systems, especially for urban rivers or lakes. However, there is little information on the water quality dynamics impacted by water exchange period.Remarks: Line 54: The choice of reference should be supplemented with respect to the limited information on pollutant degradation and its impacting mechanism, especially the interaction between water quality and soil or sediment microbial diversity, which have been developed, line 54. (e.g. Ref. Modelling water distribution network failures and deterioration, 2017, IEEE International Conference on Industrial Engineering and Engineering Management 2017-December, 924-928. DOI 10.1109/IEEM.2017.8290027; Valis, D. Perspective renewal model for water distributions systems, 26th Conference on European Safety and Reliability (ESREL), 2017, Risk, Reliability and Safety: Innovating Theory and Practice, 1050-1055. Line 376: Please provide the capital letter for the following section: 4.3. limitations of this study. What adjustable impact factors do you want to take into account in investigation the water quality dynamics and its impact mechanism. Please include this in the text. If possible, please add in the conclusions some future perspectives of work.

Author Response

I would like to express great gratitude to the reviewers and editors. 

According to the precious suggestions, we have further improved the manuscript. and the details are as follows:

1.Line 54: The choice of reference should be supplemented with respect to the limited information on pollutant degradation and its impacting mechanism, especially the interaction between water quality and soil or sediment microbial diversity, which have been developed, line 54. (e.g. Ref. Modelling water distribution network failures and deterioration, 2017, IEEE International Conference on Industrial Engineering and Engineering Management 2017-December, 924-928. DOI 10.1109/IEEM.2017.8290027; Valis, D. Perspective renewal model for water distributions systems, 26th Conference on European Safety and Reliability (ESREL), 2017, Risk, Reliability and Safety: Innovating Theory and Practice, 1050-1055. 

Re: thanks for the kind help, I have added the following references.

Valis, D.; Hasilová, K.; Forbelská, M.; Pietrucha-Urbanik, K. Modelling water distribution network failures and deterioration, IEEE International Conference on Industrial Engineering and Engineering Management 2017, 924-928. DOI 10.1109/IEEM.2017.8290027.

Valis, D. Perspective renewal model for water distributions systems, 26th Conference on European Safety and Reliability (ESREL), Reliability and Safety: Innovating Theory and Practice, 2017, 1050-1055.

2.Line 376: Please provide the capital letter for the following section: 4.3. limitations of this study.

Re: I have corrected the capital letter.

3. What adjustable impact factors do you want to take into account in investigation the water quality dynamics and its impact mechanism. Please include this in the text. If possible, please add in the conclusions some future perspectives of work.

Re: In the section 4.3, we have supplemented some discussion on the future work. " In the future, more organic pollutants and poisonous indexes, and environmental conditions should be considered that can be more sensitive to health of aquatic species in replenishment to rivers and lakes, and it would be better to conduct an on-site and long-term investigation using natural ponds or lakes." (see  line 376-379).

meanwhile, in the section of 5. conclusion, we have revised the last sentence, and the adjustable factors should include the proportion of reclaimed water and other environmental conditions.  "It is worth further investigating the water quality dynamics and its impact mechanism under the conditions combined natural water ecosystems with adjustable impact factors including the proportion of reclaimed water and its sensitive environmental conditions." (see line 399-402.)

"

Reviewer 2 Report (New Reviewer)

No further comment.

Author Response

I would like to express great gratitude to the reviewer's precious suggestions and comments.

This manuscript is a resubmission of an earlier submission. The following is a list of the peer review reports and author responses from that submission.

Round 1

Reviewer 1 Report

The manuscript concerns the important issue of “The effect of flow exchange on water quality changes in reclaimed water replenishment city lake”.

Remarks: Background: Place the question addressed in a broad context and highlight the purpose of the study. Materials and Methods: briefly describe the main methods or treatments applied. Results: summarize the article's main findings. Conclusions: indicate the main conclusions or interpretations. More detailed information about the structure of the paper should be presented at the end of the section of the introduction.

Obvious conclusions are presented in the text. What is the novelty of the presented research compared to the other existing works. The conclusions are very vague. The results should be interpreted and deliver the meanings, this section should be more focused and based on the results. Much more consistency needs to be achieved in the interplay between results and conclusions, and it needs to be strongly revised.

Recommendations at the end of the text bring anything new and are obvious.

Reviewer 2 Report

This paper is about the impact of reclaimed water replenishment on the water quality of the environmental water bodies. This paper has interesting results, however there is a lack of clarity and explanations through the manuscript. Below some specific comments:

In the abstract, L14: add (K) next to the "degradation coefficient of pollutant". Then L17 can remove "degradation coefficient" and write K only.

In Experimental apparatus section, the 3st paragraphs should move to the introduction.

L180-185: I am a bit confused here. When referring to "both reclaimed waters", are you referring to the 2 initial water resources, so 100% reclaimed water and 50/50 mixed water as explained in L101-103? If it is the case, only t0 of the reclaimed water (Fig 2 a) greatly variate. The 0 L/d of the mixed water is similar to the other mixed water circulate flow discharges. Please clarify.

How would you explain the max peak shift of a) 28 L/d reclaimed water and b) mixed water 7L/d from ~32 to ~24d?

Fig 2. I would recommend to change the plot format. It is difficult to distinguish between the 2 blues and cannot differentiate all of them if printing in B&W. Also, samples done in duplicate. Please add standard deviation.

Fig3: Does the K value should correspond to the K value of Fig2 at 40d? It does not seem like.

Fig4: Same comments than Fig 2. Spelling check needed.

Fig 5: Like Fig3, I do not understand how you obtained the K value.

L309: Review sentence.